# Roles of Splicing Factors in Hormone-Related Cancer Progression

**DOI:** 10.3390/ijms21051551

**Published:** 2020-02-25

**Authors:** Toshihiko Takeiwa, Yuichi Mitobe, Kazuhiro Ikeda, Kuniko Horie-Inoue, Satoshi Inoue

**Affiliations:** 1Division of Gene Regulation and Signal Transduction, Research Center for Genomic Medicine, Saitama Medical University, Hidaka, Saitama 350-1241, Japan; ttakeiwa@saitama-med.ac.jp (T.T.); ymitobe31@gmail.com (Y.M.); ikeda@saitama-med.ac.jp (K.I.); khorie07@saitama-med.ac.jp (K.H.-I.); 2Department of Systems Aging Science and Medicine, Tokyo Metropolitan Institute of Gerontology, Itabashi-ku, Tokyo 173-0015, Japan

**Keywords:** DBHS family proteins, SR proteins, hnRNPs, breast cancer, prostate cancer

## Abstract

Splicing of mRNA precursor (pre-mRNA) is a mechanism to generate multiple mRNA isoforms from a single pre-mRNA, and it plays an essential role in a variety of biological phenomena and diseases such as cancers. Previous studies have demonstrated that cancer-specific splicing events are involved in various aspects of cancers such as proliferation, migration and response to hormones, suggesting that splicing-targeting therapy can be promising as a new strategy for cancer treatment. In this review, we focus on the splicing regulation by RNA-binding proteins including *Drosophila behavior/human splicing* (DBHS) family proteins, serine/arginine-rich (SR) proteins and heterogeneous nuclear ribonucleoproteins (hnRNPs) in hormone-related cancers, such as breast and prostate cancers.

## 1. Introduction

Splicing of mRNA precursors (pre-mRNAs) is an essential mechanism in the posttranscriptional regulation of gene expression. Eukaryotic protein-coding genes are usually split into exons that are intervened by introns. Mature mRNAs are generated by splicing, removing the introns from pre-mRNAs. Pre-mRNA splicing is carried out by macromolecular complexes, designated as spliceosomes [1,2]. Pre-mRNA splicing often generates multiple mRNA isoforms from a single pre-mRNA through different exon/intron recognition patterns, or alternative splicing, which gives rise to the diversity of protein-coding sequences and cellular proteome [3,4]. In humans, transcripts from more than 90% of multi-exon genes undergo alternative splicing [5,6]. Furthermore, tissue- or cell-type-specific alternative splicing has been shown to be important for various biological processes such as tissue development and cell differentiation [7]. In addition, recent studies have shown that pre-mRNA splicing is also involved in various diseases including cancers. It has been revealed that the expressions and activities of splicing factors are different among various types of cancers, which result in distinct splicing patterns of multiple transcripts that contribute to particular disease pathophysiology, such as proliferation, migration and hormone responsiveness [8,9,10,11]. Therefore, the splicing factors dysregulated in cancers would be promising new targets for cancer management including prognosis, diagnosis, and therapy. In this review, we focus on the roles of splicing factors associated with hormone-related cancers, such as breast and prostate cancers.

## 2. Hormone-Related Cancers

Several cancers including breast, endometrial, ovarian, prostate, testicular and thyroid cancers are categorized as hormone-related cancers, in which cell proliferation is driven by various endogenous and exogenous hormones [12]. Among women worldwide, breast cancer is the most commonly diagnosed cancer and the leading cause of cancer deaths. For men, prostate cancer is the second commonly diagnosed cancer, following lung cancer, and the fifth leading cause of cancer deaths worldwide. According to the *GLOBOCAN 2018*, which estimates the cancer incidence and mortality produced by the International Agency for Research on Cancer (IARC), the numbers of new cases of breast and prostate cancers in 2018 are estimated as 2,088,849 and 626,679, respectively, those of deaths due to breast and prostate cancers are estimated as 1,276,106 and 358,989, respectively [13]. The majority of breast cancers are primarily sensitive to sex steroid hormones, estrogen and progesterone, and the majority of prostate cancers are sensitive to another sex hormone, androgen [14,15,16,17]. The receptors for these sex hormones, namely, estrogen, progesterone and androgen receptors (ERs, PRs, and ARs, respectively) are ligand-dependent transcription factors. With their cognate ligands, these hormone receptors dimerize and associate with DNA through their DNA-binding domains. These hormone receptors form complexes with other transcription factors and co-regulators, such as the steroid receptor coactivator (SRC)/p160 family proteins and CREB binding protein (CBP)/p300, and they contribute to the transcriptional regulation of their target genes (Figure 1) [16,17,18,19,20,21].

Since sex hormone signaling pathways play essential roles in cancer pathophysiology; the therapies targeting the hormones and their receptors, or endocrine therapies, are the mainstays of breast and prostate cancer treatment [22,23,24,25,26,27,28,29]. For breast cancer treatment, drugs that suppress estrogen signaling or estrogen production were used for endocrine therapies. To suppress estrogen-mediated ER activation, the drugs such as selective estrogen receptor modulators (SERMs) and selective estrogen receptor degraders or down-regulators (SERDs) are used. Although both SERMs and SERDs compete with estrogen, they differently regulate ER signaling (Figure 1A). SERMs modulate the ER activity by affecting the interaction with co-factors, resulting in the changes of ER-targeted gene expression. SERMs like tamoxifen and raloxifene function as ER antagonists in breast cancer and are used for breast cancer therapy or prevention. In contrast, SERDs, such as fulvestrant, induce the destabilization of ER and abolish ER signaling [23]. Aromatase inhibitors and luteinizing hormone-releasing agonists are used to prevent the estrogen synthesis [22]. For prostate cancer treatment, orchiectomy, a surgery to eliminate testis, has been originally performed as androgen deprivation therapy for years [24,25]. For chemical castration, gonadotropin-releasing hormone agonists and antagonists are used to prevent the androgen synthesis. Abiraterone has been recently used for metastatic castration-resistant prostate cancer (CRPC) as a selective inhibitor for androgen biosynthesis by blocking the enzymatic activity of cytochrome P-450c17 (CYP17) [26]. As androgen receptor antagonists that prevent androgen–AR interaction, flutamide and bicalutamide have been used for years, and recently more potent AR antagonists, such as enzalutamide, has been applied to CRPC and reported to decrease the risk of cancer progression and death [27]. Selective androgen receptor modulators (SARMs) that function as AR antagonists in specific cell types including prostate cancer cells, and selective androgen receptor degraders or down-regulators (SARDs) are being studied for clinical application (Figure 1B) [28,29].

Although the endocrine therapies are initially successful, both breast and prostate cancers eventually acquire the therapy resistance. Endocrine therapy-resistant CRPC usually arises in the majority of prostate cancer patients with treatment within two to three years [30]. On the contrary, it has been reported that the distant recurrence of breast cancer occur in 20%–50% of the patients during the period of 20 years from the start of the endocrine therapy [31]. While the majority of breast cancers are sensitive to the endocrine therapy, patients with triple-negative breast cancer (TNBC) exhibit poor outcomes as the subtype lacks effective therapeutic targets including ER, PR and human epidermal growth factor receptor 2 (HER2)/ErbB2. Thus, novel strategies for breast and prostate cancers remain to be developed. Recent studies have indicated that the cancer-specific splicing events are closely associated with cancer progression, suggesting that splicing factors could be promising targets for cancer treatment [8,9,10]. Interestingly, alternative splicing of AR in prostate cancer is suggested to be important for acquiring the endocrine therapy resistance, as described in a later part.

## 3. Pre-mRNA Splicing and Spliceosome Assembly

Pre-mRNA splicing is carried out by spliceosomes [1,2] and there are two types of spliceosomes in humans, known as major (U2-dependent) and minor (U12-dependent) spliceosomes. The major and minor spliceosomes recognize and excise the major (U2-type) or minor (U12-type) introns, respectively. The most human introns are the major introns, while the minor introns make up only ~0.35% of human introns and are contained in 700-800 genes, each of which basically has one minor intron and multiple major introns [32]. The major spliceosome consists of five small nuclear ribonucleoproteins (snRNPs) including U1, U2, U4, U5, and U6, and proteins called splicing factors. The assembly of the major spliceosomes requires the conserved sequences of the pre-mRNA, the 5’ splice site (5’ SS), the branchpoint sequence (BPS), and the 3’ splice site (3’ SS). The 3’ SS contains the continuous pyrimidine sequences termed polypyrimidine tract (PPT) and following conserved AG-dinucleotides (Figure 2A). In the early phase of the spliceosome assembly, U1 snRNP, splicing factor 1 (SF1)/branchpoint binding protein (BBP) and U2 snRNP auxiliary factor (U2AF) bind to the 5’ SS, BPS and the 3’ SS, respectively (E complex). U2AF is a heterodimer of U2AF^65^ and U2AF^35^, which recognize PPT or AG-dinucleotides of the 3’ SS, respectively. Next, U2 replaces SF1 to generate A complex. U2 snRNP consists of U2 small nuclear RNA (snRNA) and multiple proteins, including SF3b complex, and recognizes BPS through base-pairing between U2 snRNA and BPS. U4, U5 and U6 form a complex, U4/U6.U5 tri-snRNP, which is recruited to pre-mRNA to form B complex. After U1 and U4 exit, the major spliceosome becomes catalytically active (B* complex) and catalyzes two-sequential steps of splicing, resulting in ligation of exons and removal of lariat introns (Figure 2B) [1]. Although the 5’/3’ SS and BPS are required for pre-mRNA splicing, these sequences alone are often insufficient to exon/intron recognition. It has been revealed that the majority of pre-mRNAs possess the additional splicing regulatory sequences in exonic and intronic regions. The sequences that enhance the intron exclusion are termed exonic or intronic splicing enhancers (ESEs or ISEs), while the sequences that suppress the intron exclusion are called exonic or intronic splicing silencers (ESSs or ISSs) [3,4]. Splicing factors, such as serine/arginine-rich proteins (SR proteins) and heterogeneous nuclear ribonucleoproteins (hnRNPs), bind to these splicing regulatory elements and regulate the spliceosome assembly (Figure 2C). SR proteins are RNA-binding proteins characterized by having one or two RNA recognition motifs (RRMs) in their N-termini and a domain rich in arginine and serine (RS domain) in their C-termini. RS domains are involved in the interaction with other splicing factors including SR proteins [33,34]. SR proteins generally bind to ESEs and ISEs and facilitate intron exclusion through interaction with U1 and U2AF. On the other hand, hnRNPs typically bind to ESSs and ISSs to inhibit the intron exclusion. The functions of SR proteins and hnRNPs; however, are shown to be context-dependent [3,4].

In the minor spliceosomes, U11, U12, U4atac and U6atac replace U1, U2, U4 and U6, respectively. U5 snRNP is shared by both spliceosomes [32,35]. The minor introns have 5’/3’ SS and BPS, which have different sequences from those of the major introns. In contrast to the sequential recognition of the 5’ SS and BPS by U1 and U2, U11 and U12 form the U11/U12 di-snRNP complex, and cooperatively recognize these sequences. Although U2AF recognizes the 3’ SS in major introns, the 3’ SS in minor introns lacks PPT and is recognized by zinc finger CCCH-type, RNA binding motif and serine/arginine rich 2 (ZRSR2)/U2AF^35^-related protein (Urp). ZRSR2 was also shown to function in the major spliceosome, specifically required for the second step of splicing [36].

## 4. Alteration of Splicing Factors in Cancer

### 4.1. Somatic Mutations of Splicing Factors in Cancer

Recent whole-genome sequencing analyses have revealed that recurrent somatic mutations occur in some genes of splicing factors in cancers [8,37]. Mutations of splicing factors were initially identified in patients with several types of hematological malignancies, such as myelodysplastic syndromes (MDS) and related disorders (myelodysplasia) [38,39,40,41], as well as chronic lymphocytic leukemia (CLL) [42,43,44]. Especially, the majority of splicing factors that are frequently mutated in those cancers are components of the E/A complex and are involved in BPS and 3’ SS recognition, suggesting that the specific steps of the spliceosome assembly may have crucial roles in hematological malignancies. The most common mutations are found in *splicing factor 3b subunit 1* (*SF3B1*), *U2AF^35^*, *serine/arginine-rich splicing factor 2* (*SRSF2*) and *ZRSR2* [8,37]. SF3B1 is a component of SF3b complex and SRSF2 is a member of SR proteins. In fact, some mutations in *SF3B1*, *U2AF^35^* and *SRSF2* have been demonstrated to affect alternative splicing [38,40,44,45,46,47,48,49,50,51,52], and *SF3B1* and *SRSF2* mutations are suggested to be adverse prognostic risks of CLL and of MDS, respectively [43,53,54,55,56]. Interestingly, the splicing factor mutations occur in a mutually-exclusive manner in myelodysplasia [38]. It may suggest synthetic lethality of those mutations. *SF3B1* and *U2AF^35^* mutations have been also found in the solid cancers. *SF3B1* mutations have been identified in uveal melanoma (15%–29%), cutaneous melanoma (1%), pleural mesothelioma (2%), pancreatic ductal adenocarcinoma (3%), breast cancer (2%–4%), and prostate cancer (1.1%), while *U2AF^35^* mutations were found in lung adenocarcinoma (3%) and prostate cancer (0.5%) [8,57].

Furthermore, mutations occur not only in protein components of the spliceosome. Recently, mutations in an RNA component of the spliceosome were discovered in several types of cancers. U1 snRNA is an RNA component of U1 snRNP and plays an essential role in the initial recognition of 5’ SS through base-pairing between the 5’ part of U1 snRNA and 5’ SS. It was reported that several bases of U1 snRNA are mutated in some cancers including CLL, hepatocellular carcinoma (HCC) and Sonic hedgehog (SHH) medulloblastomas [58,59]. Among these mutations, mutations of the third base of U1 snRNA demonstrated the changes of 5’ SS recognition. A to C or G mutation of the third base of U1 snRNA results in mis-splicing of several cancer-related genes such as *CD44*, *Musashi RNA binding protein 2* (*MSI2*) and *DNA polymerase delta 1, catalytic subunit* (*POLD1*) in CLL and *Patched 1* (*PTCH1*) and *GLI family zinc finger 2* (*GLI2*) in SHH medulloblastomas, respectively [58,59]. In addition, these U1 snRNA mutations were suggested to be associated with the pathophysiology of CLL, HCC and SHH medulloblastoma. Thus, mutations of early spliceosome complexes are closely associated with the diverse types of cancers and initial steps of the spliceosome assembly could be promising targets for cancer treatment.

### 4.2. Deregulated Gene Expression of Splicing Factors in Cancer

Although recurrent somatic mutations of splicing factors are detected in the solid cancers, those mutations are generally rare. Instead, alteration of gene expression is frequently observed. Deregulated expression of some splicing factors such as DBHS family proteins and SR proteins induces cancer-specific splicing events and promotes cancer progression. However, whether the splicing factors function as oncogenes or tumor-suppressors is dependent on cancer types. For example, it was shown that serine/arginine-rich splicing factor 3 (SRSF3) promotes the progression of breast cancer but suppresses hepatic carcinogenesis [60,61]. Moreover, it has been implicated that the post-translational modification of splicing factors regulates cancer-specific splicing events. For instance, SR protein kinases (SRPKs) and CDC-like kinases (CLKs) phosphorylate and regulate the activity of SR proteins and control alternative splicing events [29].

In the following sections, we introduce the roles of DBHS family proteins, SR proteins and hnRNPs in breast and prostate cancers. In addition, we also discuss the functions of other splicing factors and splicing regulatory proteins in those cancers.

## 5. The Roles of Splicing Factors and Splicing Regulatory Proteins in Breast and Prostate Cancers

### 5.1. DBHS Family Proteins

The DBHS family proteins are defined by a conserved core region termed the DBHS region. The DBHS region consists of tandem RNA recognition motifs (RRM1 and RRM2), a NonA/paraspeckle (NOPS) domain and a C-terminal coiled-coil domain (Figure 3A) [62]. In humans, there are three DBHS family members: PTB-associated splicing factor (PSF)/splicing factor proline/glutamine rich (SFPQ), Non-POU domain-containing octamer-binding protein (NONO)/nuclear RNA-binding protein, 54 kDa (p54^nrb^) and paraspeckle component 1 (PSPC1)/paraspeckle protein 1 (PSP1). These proteins form homodimer or heterodimer with other DBHS family proteins by reciprocal interactions between RRM2, NOPS and coiled-coil domains [62]. In addition, a recent study suggests that oligomerization of DBHS family proteins is mediated by the interaction between coiled-coil domains [63]. The DBHS family proteins can interact with proteins, RNA and DNA, and are involved in multiple cellular events such as pre-mRNA splicing, transcriptional activation and repression, DNA repair and the formation of a subnuclear ribonucleoprotein body, or paraspeckle [62]. It has recently been reported that PSF and NONO in breast and prostate cancers promote cancer progression by regulating cancer-specific splicing events. In the following sections, we describe recent findings on splicing regulation by PSF and NONO in both cancers.

#### 5.1.1. PSF

As for *PSF* expression in prostate cancer, *PSF* expression is upregulated in prostate cancer cells compared to normal prostate epithelial cells [64]. In addition, PSF protein is highly expressed in a subset of tumor samples and higher expression of PSF correlates with cancer-specific survival after surgery and the prostate specific antigen (PSA)-free survival after endocrine therapy. Moreover, *PSF* mRNA is increased in metastatic and advanced prostate cancer clinical samples [64]. These data suggest that PSF plays important roles in the pathophysiology of prostate cancer. Consistently, PSF is demonstrated to be essential for the in vitro growth of hormone-refractory prostate cancer cells and the in vivo tumor growth in castrated mice as a CRPC model [64]. PSF binds to multiple transcripts and upregulates the stability of its target transcripts. Moreover, the main targets of PSF are transcripts of the spliceosome genes such as *SF1* and *splicing factor 3b subunit 2* (*SF3B2)*. Therefore, PSF would control a broad range of cancer-specific splicing events by upregulating the expression of spliceosome genes, resulting in the activation of oncogenic pathways (Figure 3C) [64].

Importantly, PSF targets include an AR splice variant, AR-V7, which is closely linked to the acquirement of endocrine therapy resistance [64]. Among multiple AR splice variants, AR-V7 is the most common AR variant in CRPC [65,66,67,68]. Previous studies indicated that AR-V7 expression was detected in circulating tumor cells (CTCs) from CRPC patients and associated with endocrine therapy resistance using abiraterone [26] and enzalutamide [27]. Moreover, AR-V7-positive CTCs are associated with worse PSA response rates and shorter progression-free and overall survival rates. Therefore, AR-V7 may be promising for a potential target of diagnosis, prognosis and therapy in metastatic CRPC [69,70,71]. The full-length AR protein is encoded by *AR* isoform composed of eight exons and consists of four core domains: The N-terminal domain (NTD), the DNA binding domain (DBD), the hinge region and the C-terminal ligand binding domain (LBD). In contrast, AR-V7 protein is encoded by *AR* variant composed of exons 1, 2, and 3 and a cryptic exon termed exon 3B or cryptic exon 3 (CE3), which is generated by alternative use of the cryptic 3’ SS in intron 3 (Figure 3B). The resultant AR-V7 protein includes the NTD and DBD, but lacks the hinge region and LBD [72,73]. The AR-V7 protein is constitutively active and forms a heterodimer with the full-length AR protein, leading to the androgen-independent activation of canonical AR signaling. Moreover, the AR-V7 protein is indicated to form homodimer and regulate a unique set of genes [72,73].

PSF is suggested to regulate the splicing events that produce the full-length AR and AR-V7 variant. PSF forms a complex with NONO and other splicing factors including U2AF^65^, hnRNPU and DDX23, and promotes the expression of mRNAs encoding the full-length AR and AR-V7 (Figure 3C) [64]. Moreover, PSF upregulates the expression of *SF3B2* that is a component of the SF3b complex, and associates with the splicing of *AR* pre-mRNA to generate AR-V7 variant as mentioned below [64]. In addition, transcripts of AR-regulated genes are PSF targets, suggesting that PSF plays a role in AR signaling. Thus, PSF may regulate the AR signaling pathway at multiple steps, which lead to prostate cancer progression [64].

#### 5.1.2. NONO

It is recently indicated that high-NONO immunoreactivity associates with poor prognosis of breast cancer patients [74]. Public breast cancer databases also show that *NONO* mRNA expression is increased in invasive ductal breast cancer compared to normal breast samples, and higher expression of *NONO* mRNA associates with poor distant disease-free and overall survivals of the patients with all subtypes and ER-positive subtype of breast cancers [74]. In terms of prostate cancer, the expression of *NONO* is upregulated in CRPC and metastatic prostate cancers, and increased expression of *NONO* associates with poor prognosis of prostate cancer patients [64]. These data suggest that NONO plays a key role in the progression of both breast and prostate cancers [64,74].

A recent study reported that NONO in breast cancer regulates the expression of *S-phase-associated kinase 2* (*SKP2*) and *E2F transcription factor 8* (*E2F8*) at the post-transcriptional level, promoting breast cancer proliferation (Figure 3C) [74]. In this report, NONO knockdown decreased the expression level of exonic regions of these mRNAs, while hardly affected the expression of intronic regions. These data suggest that NONO may regulate the expression of *SKP2* and *E2F8* by controlling their splicing processes [74]. As for NONO functions in prostate cancer, it is indicated that NONO promotes the proliferation of prostate cancer cells and controls the splicing regulation, which upregulates the expression of mRNAs encoding the full-length AR and AR-V7 proteins in CRPC (Figure 3C). Moreover, NONO increases the expression of some splicing factors, which would affect the splicing events in CRPC [64]. In addition, NONO modulates alternative splicing of *ephrin type-A receptor 6* (*EPHA6*) pre-mRNA in CRPC [75]. EPHA6 is an ephrin receptor reported to be overexpressed and to promote angiogenesis and metastasis in prostate cancer. NONO induces the production of a splice variant, EPHA6-001, which contributes to the growth of CRPC by unknown mechanisms [75].

Thus, recent findings have shown that the DBHS family proteins PSF and NONO regulate cancer-specific splicing events and promote breast and prostate cancer progression. Future studies will elucidate the precise molecular mechanisms of splicing regulation by these DBHS family proteins.

### 5.2. SR Proteins

In breast cancer, a member of SR proteins, SRSF1, has been shown to regulate various alternative splicing events of cancer-related genes and promote cancer progression. Recepteur d’origine nantais (Ron) is a tyrosine kinase receptor for macrophage-stimulating protein (MSP) and is involved in cell motility and matrix invasion. SRSF1 is suggested to interact with an exonic splicing enhancer (ESE) in exon 12 of *Ron* pre-mRNA and induce the skipping of exon 11 [75]. The resultant mRNA isoform is translated to a constitutively active isoform of Ron, ΔRon, which upregulates breast cancer cell motility [76]. Moreover, SRSF1 controls alternative splicing of *myeloid cell leukemia-1* (*Mcl-1*) pre-mRNA [77]. Mcl-1 is a member of the B-cell lymphoma 2 (Bcl-2) family and regulates the apoptotic process. An *Mcl-1* pre-mRNA contains three exons, and the inclusion or skipping of exon 2 results in the production of two functionally different isoforms, the pro-apoptotic Mcl-1_S_ and the anti-apoptotic Mcl-1_L_. SRSF1 is indicated to upregulate the inclusion of exon 2 to produce Mcl-1_L_ [77]. It is also suggested that SRSF1 may affect protein stability and translation of *Mcl-1* [77]. In addition to *Mcl-1*, SRSF1 controls splicing events of another Bcl-2 family member, *Bcl-2 interacting mediator of cell death* (*BIM*)/*Bcl-2 like 11* (*BCL2L11*), leading to the upregulation of splice variants, BIM γ1 and γ2 [78]. Although BIM possesses the BH3 domain that is necessary for the induction of apoptosis, BIM γ1 and γ2 lack the BH3 domain. Importantly, overexpression of BIM γ1, but not γ2, was reported to promote the growth and survival of MCF-10A mammary epithelial cell line cultured in Matrigel [78]. Therefore, splicing regulation of *Mcl-1* and *BIM* by SRSF1 may contribute to breast cancer cell survival. Furthermore, SRSF1 regulates the expression of *bridging integrator 1* (*BIN1*) and *mitogen-activated protein kinase* (*MAPK*) *signal-integrating kinase 2* (*Mnk2*), which are tumor suppressor and kinase, respectively [78]. SRSF1 promotes the inclusion of exon 12a of *BIN1* to generate an isoform, BIN+12a, which lacks tumor-suppressive activity. On the other hand, SRSF1 enhances the inclusion of exon 13b of *Mnk2*, generating the Mnk2b isoform that increases the phosphorylation of translational initiation factor eIF4E. Mnk2-mediated hyperactivation of eIF4E may promote the growth and survival of breast cancer. Thus, SRSF1 regulates various aspects of breast cancer such as cell proliferation and motility.

Meanwhile, it was reported that SRSF1 regulates splicing of *AR* pre-mRNA to generate AR-V7 in prostate cancer [79]. Moreover, SRSF1 regulates the production of cyclin D1 (CCND1) isoform CCND1b in prostate cancer [80]. It was previously reported that CCND1b was upregulated in prostate cancer compared to non-neoplastic tissue and could promote anchorage-independent growth and cell invasiveness [81,82]. A transcript encoding CCND1b is generated by the premature transcriptional termination in intron 4 and the partial retention of intron 4. CCND1b lacks exon 5-encoded amino acids and contains a novel C-terminal domain of unknown function. SRSF1 is suggested to bind to the *CCND1* exon 4–intron 4 boundary and promote CCND1b expression [80]. Since the upregulation of CCND1b was also observed in breast cancer, SRSF1 may regulate CCND1b expression in the similar manner in breast and prostate cancers [83].

In addition to SRSF1, the roles of SRSF5 in breast and prostate cancers have been reported. In breast cancer, SRSF5 is shown to regulate alternative splicing of *Mcl-1* pre-mRNA to produce Mcl-1_L_ [77]. On the other hand, SRSF5 is indicated to be involved in pre-mRNA splicing of *Kruppel-like factor 6* (*KLF6*) in prostate cancer [84,85]. KLF6 is a member of Kruppel-like zinc finger transcription factors and functions as a tumor suppressor. The allelic loss and somatic mutations of *KLF6* were observed in patients with sporadic prostate cancer and a germline single nucleotide polymorphism (SNP) in *KLF6* gene IVS1-27 G > A is shown to cause aberrant splicing to generate 3 splice variants, KLF6-SV1, -SV2 and -SV3. Moreover, KLF6-SV1 and -SV2 antagonize wild-type KLF6 function and promote cell proliferation through decreasing the expression of p21 [84,85]. It is suggested that IVS1-27 G > A polymorphism generates a novel binding site of SRSF5, which enhances alternative splicing of *KLF6* in a dose-dependent manner [84].

Furthermore, the roles of other SR proteins, including SRSF3, SRSF4 and SRSF6, have been shown in breast cancer. It was reported that SRSF3, in cooperation with another splicing factor TAR DNA-binding protein 43 (TDP43), plays an important role in the growth and metastasis of TNBC tumors. SRSF3 and TDP43 enhance the skipping of exon 12 of *partitioning defective 3* (*PAR3*) and the inclusion of exon 12 of *NUMB*, and these splicing alterations promote the growth and metastasis of TNBC tumors, respectively [60]. In addition, SRSF3 is indicated to regulate alternative splicing of *HER2* [86]. SRSF3 enhances the removal of intron 15 and the skipping of exon 16 of *HER2*. These changes in *HER2* splicing downregulate the p100 isoform and upregulate the Δ16HER2 isoform, the former is a secreted protein that inhibits cell proliferation and the latter is a constitutively active isoform, respectively. Therefore, SRSF3 may contribute to cell proliferation through modulating the production of *HER2* isoforms [86].

SRSF4 and SRSF6, like SRSF1, were reported to play crucial roles in the proliferation of MCF-10A cells cultured in Matrigel [87]. SRSF4 and SRSF6 controlled alternative splicing events of genes that regulate transformation-associated processes and the expression of genes associated with cell proliferation, migration, cytoskeleton organization, polarity, cell signaling or cholesterol metabolism [87]. In addition, SRSF4 is suggested to be involved in cell death caused by cisplatin treatment possibly through the splicing regulation [88]. Thus, SR proteins regulate a broad range of cellular processes to promote breast and prostate cancer progression by controlling cancer-specific splicing events, suggesting that the expressions and activities of SR proteins are important for cancer progression.

### 5.3. hnRNPs

Previous studies suggested that hnRNPs play key roles in the production of AR-V7 in prostate cancer. A member of hnRNPs, hnRNPF, is shown to enhance the generation of AR-V7. It was reported that histone demethylase jumonji domain containing 1A (JMJD1A) recruits hnRNPF onto the region around the 5’ side of *AR* exon 3B, and JMJD1A and hnRNPF support the recruitment of splicing factors including U2AF^65^ to the 3’ SS in front of exon 3B, which leads to the inclusion of exon 3B to produce AR-V7 [89]. Moreover, another hnRNP family member hnRNPA1 is suggested to be involved in AR-V7 expression [90,91,92]. It was previously reported that hnRNPA1 is highly expressed in prostate tumors, compared to benign prostates, and upregulates the expression of AR-V7 [90]. Furthermore, it was shown that the downregulation of hnRNPA1 by quercetin, a flavonoid abundantly present in plants, concomitantly decreases the expression of AR-V7, resensitizing enzalutamide-resistant prostate cancer cells to enzalutamide treatment in mouse xenografts [91]. However, another study suggested that hnRNPA1 suppresses AR-V7 expression in prostate cancer cells [92]. This contradiction may be due to the difference of cell lines used in their studies, suggesting cell-type specific functions of hnRNPA1 in splicing regulation. In addition, hnRNPL is indicated to bind to intron 3 of *AR* and to control the inclusion of another cryptic exon termed exon 2b [93]. Although the significance of exon 2b inclusion is unknown, hnRNPL regulates alternative splicing of multiple genes and circular RNA formation, and promotes prostate cancer growth and survival [93,94].

In breast cancer, hnRNPs promote cell survival and tumor metastasis. It is suggested that hnRNPF, hnRNPH1 and hnRNPK enhance the production of Mcl-1_L_ to inhibit cell death [95]. Moreover, hnRNPM was shown to regulate alternative splicing of *CD44* [96]. CD44 is a cell surface adhesion receptor involved in epithelial to mesenchymal transition (EMT) processes [97], and hnRNPM upregulates alternative splicing of *CD44*, which results in EMT and metastasis at final stage [96]. In TNBC, it was reported that a substitution mutation M276I in a chromatin remodeling protein microrchidia family CW-type zinc finger 2 (MORC2) that promotes the interaction between MORC2 and hnRNPM and the hnRNPM-mediated *CD44* splicing switch, enhancing invasion and metastasis of TNBC cells in immunodeficient mice [98].

### 5.4. Other Splicing Factors and Splicing Regulatory Proteins in Breast and Prostate Cancers

#### 5.4.1. SF3b Complex

As mentioned above, the SF3b complex is a component of U2 snRNP and the mutations of the SF3b complex component *SF3B1* have been reported in some cancers, including breast and prostate cancers [8,57]. Although less frequently than *SF3B1*, the mutations of *SF3B3* and *SF3B4*, other components of the SF3b complex, are also observed in breast cancers (0.2% each) [99]. Regarding expressions of SF3b complex components, it was reported that *SF3B1* and *SF3B3* are upregulated in fulvestrant/tamoxifen-resistant breast cancer cells, and *SF3B3* overexpression associates with poor relapse-free and overall survival in ER-positive breast cancer patients [100]. In addition, the elevated expression of another component of SF3b complex, *PHD finger 5A* (*PHF5A*)/*SF3B7,* is observed in breast tumors, and high *PHF5A* expression correlates with poor disease-free survival in breast cancer patients [101]. PHF5A is reported to upregulate the stability of SF3b complex and to modulate alternative splicing events of multiple genes involved in apoptotic processes, such as *FAS-activated serine/threonine kinase* (*FASTK*) that inhibits apoptosis [101]. These data suggest that SF3b complex may be a potential target of breast cancer treatment. On the other hand, another component of SF3b complex *SF3B2* is increased in CRPC and prostate cancers with high expression of AR-V7 and associates with poor progression-free survival in prostate cancer patients [102]. In prostate cancer, SF3B2 directly binds to CE3 of *AR* and controls alternative splicing of *AR* to produce AR-V7. Interestingly, pladienolide B, an mRNA splicing inhibitor/modulator, decreases AR-V7 expression through modulating the SF3b complex and causes the regression of prostate cancer xenografts in castrated mice. Moreover, prostate cancer xenografts with *SF3B2*-overexpression are more sensitive to pladienolide B treatment [102]. *SF3B1* is overexpressed in prostate tumors and associates with aggressive features of prostate cancers [103]. *SF3B1* expression positively correlates with *AR-V7* expression, and pladienolide B treatment decreases *AR-V7* expression. In addition, pladienolide B treatment decreases the expression of *In1-Ghrelin*, an oncogenic splice variant of *Ghrelin*, suggesting that SF3B1 or the SF3b complex associates with alternative splicing of *Ghrelin*. Supporting these findings, pladienolide B treatment is indicated to suppress the proliferation, survival and migration of prostate cancer cells [103]. Therefore, *SF3B2*, *SF3B1* and the SF3b complex may be good candidates of therapeutic targets, and small chemicals inhibiting/modulating the SF3b complex might be promising for prostate cancer treatment.

#### 5.4.2. Src Associated in Mitosis of 68 kDa (Sam68)

Sam68 belongs to the signal transduction and activation of RNA metabolism (STAR) family of RNA-binding proteins. Sam68 can interact with multiple RNAs and proteins, and is involved in various steps of gene expression, such as transcription, pre-mRNA splicing and translation [104]. Previous studies demonstrated that Sam68 plays a crucial role in alternative splicing of cancer-related genes in breast and prostate cancers. For example, Sam68 enhances the inclusion of variable exon 5 of *CD44*, promoting the migration of both cancer cells [105,106]. In prostate cancer, transcriptional coactivator staphylococcal nuclease and tudor domain containing 1 (SND1) promotes the recruitment of Sam68 onto *CD44* pre-mRNA, suggesting the significance of transcription-coupled RNA splicing events in cancer [106]. In addition, Sam68 upregulates AR-V7 and CCND1b expression by controlling alternative splicing of *AR* and *CCND1* [107,108]. Besides splicing regulation, Sam68 is shown to promote the transcription of AR-V7 target genes possibly through the interaction between Sam68 and AR-V7 [108]. Although these Sam68 functions result in cancer progression, Sam68 upregulation can cause prostate cancer cell death partly through the splicing alteration of *Bcl-2* family member *BCL-X*. There are two BCL-X isoforms generated by the alternative use of two 5’ SS regions in exon 2: anti-apoptotic BCL-X_L_ and pro-apoptotic BCL-X_S_. Sam68 controls a splicing switch toward *BCL-X_S_*, leading to apoptosis [109]. Since Sam68 can regulate cancer progression positively or negatively, it is suggested that Sam68 activity may be strictly regulated in cancer. Previous studies demonstrated that post-translational modifications play essential roles in the regulation of Sam68 activity. Especially, the acetylation of Sam68 is suggested to enhance its activities of RNA binding and splicing regulation in breast and prostate cancers. It is indicated that an acetyltransferase CBP could regulate Sam68 acetylation [110], while histone deacetylase 6 (HDAC6) causes Sam68 deacetylation in cooperation with a nuclear matrix-associated protein, scaffold/matrix-associated region-binding protein 1 (SMAR1), in breast cancer [111]. With respect to HDAC6, it is suggested that HDAC6 interacts with ERα in an ER ligand-dependent manner and subsequently upregulates deacetylation of α-tubulin in breast cancer cells [112]. The deacetylation of α-tubulin is supposed to contribute to the migration of breast cancer cells and the growth of breast cancer xenografts in nude mice [112]. Thus, the HDAC6-mediated deacetylation process might play a pivotal role in controlling breast cancer progression.

On the other hand, in prostate cancer, a factor that binds to inducer of short transcripts protein 1 (FBI-1) is shown to interact with Sam68, and control Sam68-mediated alternative splicing of *BCL-X* through histone deacetylases, suggesting the importance of acetylation for Sam68 activity [109]. Targeting these Sam68 acetylation mechanisms may provide a novel therapeutic approach in breast and prostate cancers.

#### 5.4.3. SR Protein Kinases

SR proteins are phosphorylated by multiple kinases including SRPKs and CDC-like kinases (CLKs). Phosphorylation of SR proteins is essential for their nuclear import and interactions with target RNAs and other splicing factors [34]. Considering the roles of SR proteins in cancer-specific splicing events, kinases that phosphorylate SR proteins could be promising targets for cancer treatment. SRPK1, a member of SRPK, is shown to be highly expressed in various cancers including breast and prostate cancers [113,114,115,116]. Among breast cancer cells, ER-negative breast cancer cells show higher expression of SRPK1, and knockdown of SRPK1 was reported to inhibit metastasis of ER-negative breast tumors in immunodeficient mice. Moreover, high expression of SRPK1 correlates with decreased metastasis-free survival of the patients with breast cancer and preferential metastasis to lung and brain [116]. These data suggest that SRPK1 play essential roles in breast tumor metastasis. Meanwhile, SRPK1 in prostate cancer is shown to promote angiogenesis [117]. Vascular endothelial growth factor (VEGF) is a key player in angiogenesis, and alternative splicing of *VEGFA* pre-mRNA generates two groups of VEGF isoforms by alternative use of two 3’ SS regions in exon 8 of *VEGFA*. The VEGF_165_ angiogenic isoform is generated by splicing using the proximal 3’ SS, while the VEGF_165_b anti-angiogenic isoform is produced by splicing using the distal 3’ SS. SRPK1 is indicated to promote the use of the proximal 3’ SS to upregulate VEGF_165_ expression, which results in angiogenesis and prostate tumor growth in nude mice [117]. Since alternative splicing of *VEGFA* was reported to be regulated by SR proteins such as SRSF1 and SRSF6 [118], SRPK1 may associate with the alternative splicing of *VEGFA* by controlling the phosphorylation states of these SR proteins. Moreover, SRPK1 is indicated to induce the phosphorylation of a splicing factor other than SR proteins. It was reported that elevated SRPK1 in breast cancer induces RNA binding motif 4 (RBM4) phosphorylation [119]. This phosphorylation causes the cytoplasmic accumulation of RBM4 and prevents RBM4-mediated alternative splicing of *Mcl-1* and *insulin receptor* (*IR*). It is shown that RBM4 upregulates Mcl-1_S_ expression and enhances the inclusion of exon 11 of *IR* to generate the IR-B pro-apoptotic isoform in normal tissue, while breast cancer cells decrease expression of Mcl-1_S_ and IR-B through enhanced the phosphorylation of RBM4, acquiring apoptotic resistance [119]. In addition to SRPK1, another member of SRPKs, SRPK2, is suggested to promote the growth and metastasis of prostate cancer, although SRPK2 targets associated with these phenomena have not been identified [120].

Moreover, CLKs is demonstrated to regulate breast cancer progression. Previous studies using small molecule inhibitors of CLKs indicated that CLKs regulate global splicing patterns in breast cancer cells and promote breast cancer cell proliferation [121,122,123]. Alternative splicing of a tumor suppressor, *p53*, is regulated by CLKs, and inhibition of CLKs by a benzothiazole compound, TG003, causes the inclusion of exon 9β or exons 9β and 9γ to generate p53β and p53γ isoforms, respectively [121]. The expression of these isoforms is lost in about 60% of breast tumors [121]. A CLK-target, SRSF1 is involved in these splicing events, suggesting the importance of CLKs-SRSF1 in expression of p53 isoforms. p53β and p53γ activate the transcriptional activity of p53α isoform that is encoded by *p53* mRNA excluding exons 9β and 9γ, and suppress the growth of breast cancer cells under the normal condition. However, they enhance the growth of TG003-treated breast cancer cells [121]. Although the mechanism of the dual functions of p53β and p53γ remains to be revealed, this result implies that splicing regulation of *p53* may provide a new therapeutic strategy for breast cancer [121]. RNA interference-mediated silencing experiments indicated that a member of CLKs, CLK2, promotes cell proliferation, migration and invasion of breast cancer cells, and the growth of tumor xenograft in mice [124]. Silencing of CLK2 changes alternative splicing patterns of several EMT-related genes including enabled homolog (*ENAH*). This may be a part of the mechanism by which CLK2 regulates breast cancer progression [124].

## 6. Conclusions

It has become clear that somatic mutations and changes in expression levels of splicing factors play important roles in cancer pathophysiology. In this review, we focused on the roles of splicing factors and splicing regulatory proteins in breast and prostate cancers (Table 1). Accumulating evidence suggests that inhibition or modulation of these factors could be new therapies for breast and prostate cancers. For example, siRNAs and antisense oligonucleotides against mRNAs encoding these factors may be clinically applicable. Moreover, small molecule inhibitors of SRPKs and CLKs have been developed and were shown to inhibit cancer cell proliferation and promote apoptosis [125]. Interestingly, it was reported that intraperitoneal administration of SRPK inhibitor SPHINX suppresses the growth of prostate cancer xenografts in nude mice [117]. In addition, CLK1 inhibitor TG-003 and inhibitors of SRPKs and CLKs, called Cpd-1, Cpd-2 and Cpd-3, are shown to suppress the proliferation of breast cancer cells [121,122]. Thus, small molecule inhibitors of SRPKs and CLKs could be promising for the therapies of those cancers. In addition, small compounds inhibiting/modulating the SF3b complex might be potential candidates as therapeutic drugs for those cancers [100,101,102,103]. Further studies on cancer-specific splicing regulation will provide new therapeutic targets and advance the development of new drugs.

## Figures and Tables

**Figure 1 ijms-21-01551-f001:**
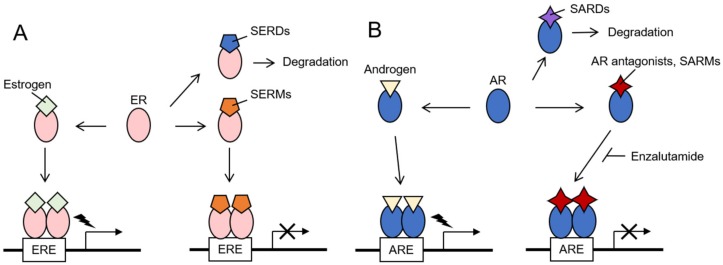
Endocrine therapy targeting estrogen and androgen receptors. (**A**) Estrogen receptor (ER) dimerizes and forms a complex with co-regulators after binding to estrogen, leading to the transcriptional regulation of ER target genes. Selective estrogen receptor modulators (SERMs) and selective estrogen receptor degraders or down-regulators (SERDs) are utilized in endocrine therapy. SERMs modulate ER activity, whereas SERDs mediate the degradation of ER, abolishing ER signaling pathway. (**B**) Androgen binding and subsequent androgen receptor (AR)-mediated transcriptional regulation. AR antagonists and selective androgen receptor modulators (SARMs) compete with androgen and modulate AR activity. Selective androgen receptor degraders or down-regulators (SARDs) promote AR degradation. ERE—estrogen response element. ARE—androgen response element.

**Figure 2 ijms-21-01551-f002:**
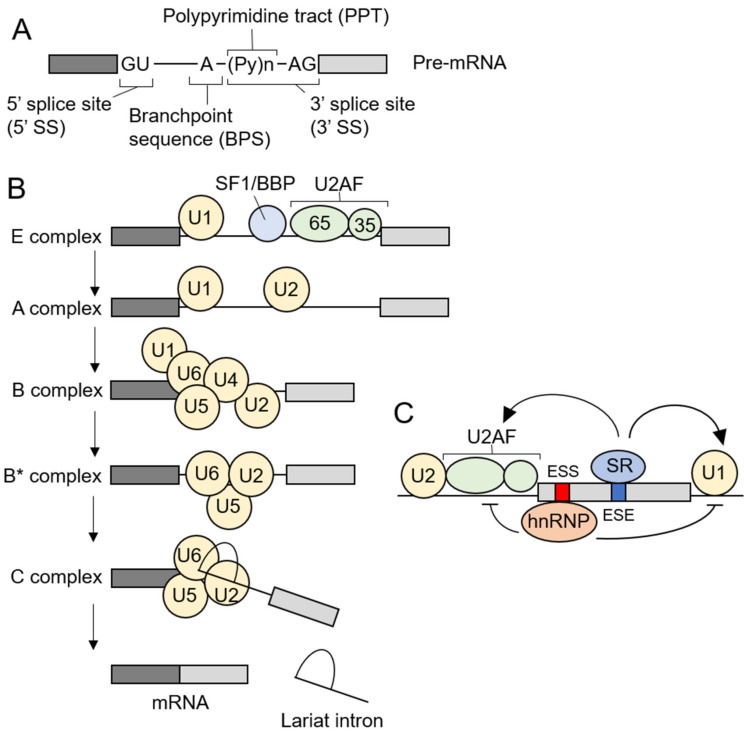
Pre-mRNA splicing mediated by the major spliceosome. (**A**) The 5 ‘splice site (5’ SS), the branchpoint sequence (BPS) and the 3 ‘splice site (3’ SS) are required for the major spliceosome assembly. The 3’ SS contains polypyrimidine tract (PPT) and AG-dinucleotides. (**B**) Schema of the major spliceosome assembly. Each step of major spliceosome assembly is described in the text (3. Pre-mRNA splicing and spliceosome assembly). (**C**) A model of splicing regulation mediated by SR proteins and hnRNPs. SR proteins usually bind to splicing enhancers and support the recruitment of splicing factors such as U1 snRNP and U2AF. hnRNPs typically bind to splicing silencers and exert the effects opposite those of SR proteins. Pre-mRNA—mRNA precursor. U2AF—U2 snRNP auxiliary factor. SF1/BBP—splicing factor 1/branchpoint binding protein. ESE—exonic splicing enhancer. ESS—exonic splicing silencer. SR proteins—serine/arginine-rich proteins. hnRNPs—heterogeneous nuclear ribonucleoproteins.

**Figure 3 ijms-21-01551-f003:**
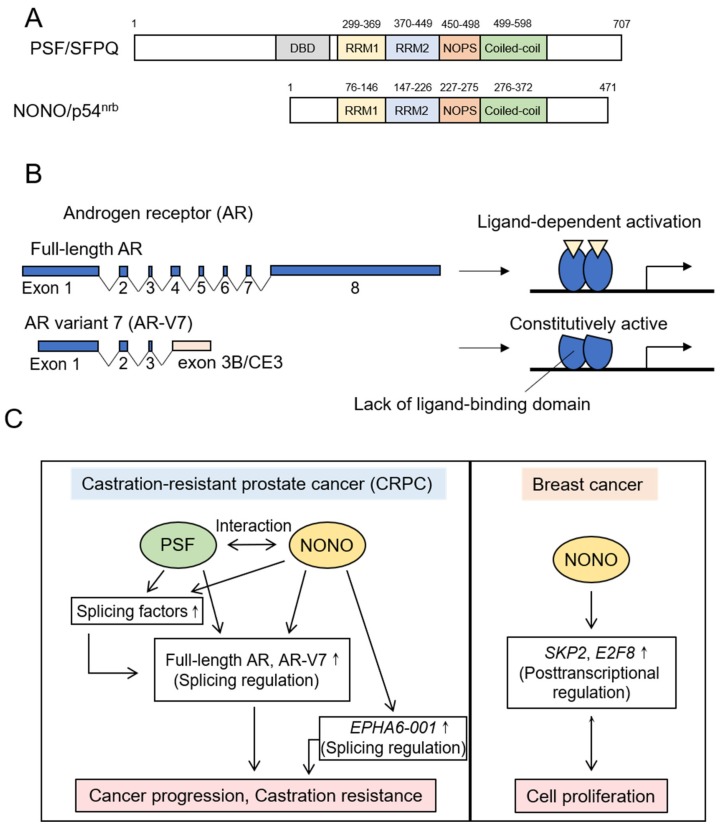
**(A**) Representation of domain structures of PTB-associated splicing factor (PSF)/ splicing factor proline/glutamine rich (SFPQ) and Non-POU domain-containing octamer-binding protein (NONO)/ nuclear RNA-binding protein, 54 kDa (p54^nrb^). PSF has the putative DNA-binding domain (DBD) N-terminal to RRM1 [62]. (**B**) Genomic structure of androgen receptor (*AR*) and alternative splicing of *AR* generating the AR-variant 7 (*AR-V7*) isoform. AR-V7 is a constitutively active form of AR that lacks its ligand-binding domain. (**C**) PSF and NONO function in prostate and breast cancers. PSF and NONO upregulate the expression of splicing factors and control the splicing of *AR* pre-mRNA to produce full-length AR and AR-V7 proteins, both of which promote castration-resistant prostate cancer (CRPC) development. NONO also modulates alternative splicing of *ephrin type-A receptor 6* (*EPHA6*) in CRPC. In breast cancer, NONO regulates the expression of *S-phase-associated kinase 2* (*SKP2*) and *E2F transcription factor 8* (*E2F8*) through posttranscriptional mechanism, leading to the promotion of cell proliferation. DBD—DNA-binding domain. RRM—RNA recognition motif. NOPS domain—NonA/paraspeckle domain.

**Table 1 ijms-21-01551-t001:** The functions of splicing factors and splicing-regulatory proteins in breast and prostate cancers.

Cancer Type	Splicing Factor	Target Genes	Effects on Cancer Progression	References
Breast cancer	NONO/p54^nrb^	*SKP2* and *E2F8*	Upregulating cell proliferation	[74]
	SRSF1	*Ron*	Activating cell migration	[76]
		*BIM*, *BIN1*	Upregulating cell proliferation Inhibiting apoptosis	[78]
		*Mnk2*		[78]
		*Mcl-1*		[77]
	SRSF3	*PAR3*	Activating cell migration and invasion	[60]
		*NUMB*	Upregulate cell proliferation	[60]
		*HER2*	Possibly upregulating cell proliferation	[86]
	SRSF4	Cisplatin-induced splicing events including *hnRNPDL* and *AMZ2*	Cisplatin-induced cell death	[88]
	SRSF4, 6	Transformation-associated genes	Activating cell migration and invasion	[87]
	hnRNPF, H1, K	*Mcl-1*	Inhibiting apoptosis	[95]
	hnRNPH1	*HER2*	Possibly upregulating cell proliferation	[86]
	hnRNPM	*CD44*	Upregulating EMT and tumor metastasis	[96,98]
	SF3B3		Associating with poor relapse-free and overall survival in ER-positive breast cancer patients	[100]
	PHF5A/SF3B7	Multiple genes involved in apoptosis including *FASTK*	Inhibiting apoptosis Associating with poor disease-free survival in breast cancer patients	[101]
	Sam68	*CD44*	Upregulating tumor metastasis	[105]
	SRPK1	*Mcl-1* and *IR*	Inhibiting apoptosis	[119]
	CLKs	*p53*	Regulating cell proliferation	[121]
	CLK2	EMT-related genes including *ENAH*	Suppressing cell migration and invasion Upregulating cell proliferation	[123]
Prostate Cancer	PSF/SFPQ	*AR*, spliceosome genes	Upregulating cell proliferation Promoting CRPC progression	[64]
	NONO/p54^nrb^	*AR*, spliceosome genes, *EPHA6*	Upregulating cell proliferation Promoting CRPC progression	[64,75]
	SRSF1	*AR*		[79]
		*CCND1*		[80]
	SRSF5	KLF6	Upregulating cell proliferation, migration, and invasion	[84,85]
	hnRNPA1	*AR*		[90,91,92]
	hnRNPF	*AR*	Possibly upregulating cell proliferation and inhibiting apoptosis	[89]
	hnRNPL	Various genes including *AR* and *MYH10*	Upregulating cell proliferation Inhibiting apoptosis	[93,94]
	SF3B1	*AR*, *Ghrelin*	Upregulating cell proliferation, survival, migration	[103]
	SF3B2	*AR*	Promoting CRPC progression	[102]
	Sam68	*CD44*		[106]
		*CCND1*		[107]
		*AR*		[108]
		*BCL-X*	Promoting apoptosis	[109]
	SRPK1	*VEGF-A*	Upregulating tumor growth via angiogenesis	[117]

*hnRNPDL*—*hnRNPD-like*. *AMZ2*—*archaelysin family metallopeptidase 2*. *MYH10*—*Myosin Heavy Chain 10*. EMT—epithelial to mesenchymal transition. CRPC—castration-resistant prostate cancer.

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
