# Peer review of "Roles of Splicing Factors in Hormone-Related Cancer Progression"

_ijms, 2020, doi:10.3390/ijms21051551_

Round 1

Reviewer 1 Report

This is a nice comprehensive review of splicing factor alterations in hormone dependent cancers.  Many reviews have focused on alterations in alternative splicing in cancer without any discussion of why splicing is changed.  This review fills that niche nicely by providing a discussion of the changes in splicing factors rather than target genes.  It also does not focus solely on SR proteins and hnRNPs, which are the two categories that are generally mentioned.  So overall I think this is a nice review that is worth publishing.  There are a few minor grammatical errors that can be corrected at the editorial level.

Reviewer 2 Report

The review submitted by Santoshi Inoue and co-workers describes the progress of knowledge in the splicing factors in the hormone related cancer progression. The article is very interesting and well-focused. Contains a number of valuable information well summarizing previous and current studies (more than 84% of cited papers were published during last 10 years, and almost 60% in the last 5 years). The references are properly chosen.

The introduction is coherent and well brings in to the topic of the paper. The Authors give a brief overview of hormone-related cancer biology, giving a special attention to the sex steroid hormones and their signalling pathways in anticancer therapy.

The review in very well organized. The content of the article matches to the profile of the journal, and well meets the formula of the review paper. Included table and figures properly  illustrate the discussed topic and increase the quality and clarity of the paper. The section ‘Conclusions’ well indicates the next potential studies directions.  

(Some minor typing errors should be eliminated)

I fully recommend the publication of the paper in the International Journal of Molecular Sciences in the present form.